# RNA-Seq and ATAC-Seq Reveal CYP26A1-Mediated Regulation of Retinoic Acid-Induced Meiosis in Chicken Primordial Germ Cells

**DOI:** 10.3390/ani15010023

**Published:** 2024-12-25

**Authors:** Zhaochuan Wang, Jiayi Chen, Jintian Wen, Siyu Zhang, Yantao Li, Jiali Wang, Zhenhui Li

**Affiliations:** 1State Key Laboratory of Swine and Poultry Breeding Industry, Guangdong Provincial Key Lab of Agro-Animal Genomics and Molecular Breeding, College of Animal Science, South China Agricultural University, Guangzhou 510642, China; zuisc@stu.scau.edu.cn (Z.W.); 13531122833@163.com (J.C.); wenjintian@stu.scau.edu.cn (J.W.); agctu.zhang@stu.scau.edu.cn (S.Z.); 20222024011@stu.scau.edu.cn (Y.L.); 15713207180@163.com (J.W.); 2Key Laboratory of Chicken Genetics, Breeding and Reproduction, Ministry of Agriculture and Rural Affair, South China Agricultural University, Guangzhou 510642, China

**Keywords:** retinoic acid, chicken primordial germ cell, meiosis, RNA-seq, ATAC-seq

## Abstract

This study investigates how retinoic acid (RA), a key signaling molecule, influences the development of chicken primordial germ cells (PGC), which are precursors to sperm and eggs. The research aimed to understand how RA helps PGC transition from an immature, proliferative state to a differentiated, more mature state, eventually entering meiosis—a crucial step in germ cell development. Using cultured chicken PGC, the study found that RA changes their cell cycle, increasing the proportion of cells in the G1 phase. Some RA-treated cells even showed early signs of meiosis. Analysis of gene activity and chromatin structure revealed that RA strongly activates *CYP26A1*, a gene responsible for breaking down RA. This suggests PGC can self-regulate RA levels to delay premature meiosis, allowing proper maturation. Other identified genes, such as *HNF1B* and *MYO1C*, may also contribute to PGC differentiation. These findings provide new insights into how RA influences germ cell development and highlight potential molecular targets for further research.

## 1. Introduction

Chicken primordial germ cells (PGC) are critical precursors of germ cells during early embryonic development, ultimately differentiating into functional gametes. Understanding the origin, characteristics, and applications of PGC is of significant importance in the fields of reproductive biology and genetics. PGC are first generated in the ectoderm and initially localized to the zona pellucida inner plate in the X-stage embryo [1]. According to the research by Hamburger and Hamilton, these cells are subsequently transferred to the germinal crescent of the embryo. During stage 4, they migrate to the germinal crescent, begin circulating in the bloodstream during stages 10–12, and enter the primordial gonads during stages 20–24, where they then commence differentiation into either male or female gametes [2,3,4].

Research has demonstrated that retinoic acid (RA), secreted by mesonephric-associated mesenchymal cells, plays a critical regulatory role in triggering meiosis in PGC during embryonic development. In mouse embryonic ovaries, high levels of RA induce the expression of *STRA8*, a key gene that initiates meiosis, thus promoting PGC to enter meiosis [5,6]. In contrast, in embryonic testes, RA is metabolized by the enzyme *CYP26B1*, which is expressed by somatic cells of the testis, preventing PGC from initiating meiosis. Only after sexual maturation, when the expression of *CYP26B1* declines, do PGC in the testes respond to RA signaling and begin meiosis [7]. In RA-deficient mouse ovaries, most PGC fail to enter meiosis, remaining undifferentiated, but in normal ovaries, approximately 75% of PGC successfully undergo meiosis [8]. These findings demonstrate the essential role of RA in initiating meiosis in PGC. Additionally, supplementing RA in vitro cultures of mouse embryonic testes compensates for the loss of endogenous RA, thereby inducing germ cells to enter meiosis [6,7]. The levels of RA within tissues are tightly regulated through a balance between its synthesis by retinaldehyde dehydrogenase and its oxidative degradation by cytochrome P450 enzymes [9]. The *CYP26A1*, *CYP26B1*, and *CYP26C1* genes encode three cytochrome P450 enzymes involved in this degradation process [10,11]. In mouse RA signaling, *CYP26B1* plays a pivotal role. In *CYP26B1*-deficient mice, elevated expression of meiosis-initiating genes such as *STRA8* and *SCP3* has been observed in the testes, alongside a threefold increase in RA levels [12]. *CYP26A1* has been demonstrated to be crucial for regulating RA gradients during zebrafish embryonic development [13]. *CYP26C1*, although less studied, is also capable of modulating specific developmental processes by controlling local RA concentrations [14].

Although RA has been established as an effective inducer of meiosis, its mechanism of action on gonadal cells remains unclear. To investigate the regulatory mechanisms of RA signaling, this study utilizes a feeder-free culture environment for chicken PGC as a model to examine whether exogenous RA can directly influence chicken PGC and induce their entry into meiosis. ATAC-seq is a method for assessing chromatin accessibility, revealing potential regulatory elements like enhancers and promoters. RNA-seq, on the other hand, quantifies gene expression levels and captures transcriptomic changes. By combining these approaches, we aim to link chromatin accessibility changes with transcriptional regulation, providing a comprehensive understanding of how RA influences PGC differentiation.

## 2. Materials and Methods

### 2.1. Materials and Reagents

Retinoic acid was purchased from Sigma-Aldrich Sigma-Aldrich (302-79-4, St. Louis, MO, USA). The cell DNA extraction kit was sourced from Omega Bio-tek (D3396, Norcross, GA, USA); RNA extraction kits were obtained from Magen Biotech (R4114, Guangzhou, China); quantitative polymerases (Q711, Nanjing, China) and reverse transcriptase (R333, Nanjing, China) were sourced from Vazyme Biotech; DMEM/F12 medium was acquired from Gibco (C11330500BT, Grand Island, NY, USA). Mouse anti-SSEA-1 (AB_528475, Iowa City, IA, USA) and EMA-1 (AB_531885, Iowa City, IA, USA) antibodies were purchased from DSHB; propidium iodide (C1052, Shanghai, China) and an alkaline phosphatase assay kit (C3206, Shanghai, China) were obtained from Beyotime Biotechnology; and a Periodic Acid Schiff stain kit and Giemsa staining solution were purchased from Solarbio (G1281, Beijing, China).

### 2.2. Isolation and Cultivation of Chicken PGC

Eggs from Zhongshan Shalan chickens were incubated at 37 °C with 60% relative humidity. At embryonic day 7.5, chicken embryos were removed using forceps and placed ventral side up. After opening the abdominal cavity, the viscera were gently displaced to expose the gonads on the renal tissue. The left and right gonads were carefully excised with forceps, washed in PBS (Gibco, C10010500BT, Grand Island, NY, USA), and transferred into centrifuge tube. The gonadal tissues were treated with 0.05% trypsin-ethylenediaminetetraacetic acid (Gibco, 25300054, Grand Island, NY, USA) for 10 min at 37 °C, and the digestion was terminated using The Dulbecco’s Modified Eagle Medium/Nutrient Mixture F-12 (DMEM/F12) containing 10% fetal bovine serum. After 4–5 h of incubation, the supernatants were aspirated and centrifuged to collect the precipitate containing the cells. The cells were resuspended in knockout DMEM and inoculated in inset-containing 6-well plates with mouse embryonic fibroblasts (MEF) at the bottom [15].

### 2.3. Preparation of Feeder Cells

After primary PGC isolation, co-culture with mouse embryonic fibroblast is essential to maintain the survival and proliferation of PGC. MEF were seeded onto a 6-well plate and allowed to proliferate until they reached over 90% confluence. The MEF were then treated with mitomycin C (Selleck, S8146, Houston, TX, USA) for 2.5 h to inhibit further proliferation. After treatment, the cells were washed three times with PBS, followed by the addition of fresh culture medium. The MEF were incubated for an additional 48 h before being used as feeder cells.

### 2.4. Exogenous RA Administration to PGC

All-trans RA was diluted in dimethyl sulfoxide (Sigma-Aldrich, 67-68-5, St. Louis, MO, USA) and then prepared in complete PGC culture medium to make a final concentration of 1µM for use. PGC were collected by centrifugation and divided into two tubes, with each being resuspended in either standard complete medium or complete medium containing 1µM RA. Cells were incubated in a 24-well plate for 1 day before being harvested for subsequent experiments.

### 2.5. RNA Extraction and Real-Time RT-PCR

The total RNA was extracted from RA-treated and untreated PGC for quantitative real-time PCR analysis(qRT-PCR). Complementary DNA (cDNA) was synthesized using reverse transcriptase and primers. The PCR amplification was performed using the synthesized cDNA as a template under the following conditions: an initial denaturation at 95 °C for 5 min, followed by 40 cycles of 95 °C for 30 s, 60 °C for 30 s, and 72 °C for 1 min, with a final extension at 72 °C for 10 min. Primer sequences were designed based on the gene sequences available on NCBI, and Primer 5.0 software was used for primer design.

### 2.6. Immunofluorescence Staining of PGC

Male and female PGC were collected separately, washed twice with PBS, and fixed with 4% paraformaldehyde (Beyotime, P0099, Shanghai, China) for 30 min. The cell suspension was then transferred onto pre-treated glass slides and allowed to air dry. Next, 5% BSA (Beyotime, P0102, Shanghai, China) was added to the slides and incubated at room temperature for 30 min; excess solution was removed without washing. Primary antibodies, mouse anti-SSEA-1 (DSHB, AB_528475, Iowa City, IA, USA), and EMA-1 (DSHB, AB_531885, Iowa City, IA, USA) were added to the slides and incubated overnight at 4 °C, followed by three 5-min washes with PBS. Fluorescently labeled secondary antibodies (Goat Anti-Mouse lgG, Abbkine, A23210, Wuhan, China) were applied, and samples were incubated at 37 °C for 30 min in the dark, followed by three additional 5-min washes with PBS. DAPI staining solution (Beyotime, P0131, Shanghai, China) was then added, and slides were incubated in the dark at 37 °C for 5 min, followed by two 5-min PBS washes. Finally, an anti-fade mounting medium was applied to the slides for observation under a fluorescence microscope, and images were captured.

### 2.7. AP and PAS Staining of PGC

Male and female PGC were collected separately. For alkaline phosphatase (AP) staining, cells were washed twice with PBS, resuspended in pre-prepared NBT/BCIP staining solution (C3206, Shanghai, China), and incubated in a 24-well plate for 2 h. AP activity is a key indicator of stem cell pluripotency; as PGC are a type of stem cell, they stain dark red with AP staining. For Periodic Acid–Schiff (PAS) staining, after collecting PGC, cells were washed twice with PBS and placed onto slides, followed by fixation in 4% paraformaldehyde for 30 min. After air drying, cells were oxidized with Periodic Acid and stained with Schiff reagent (G1281, Beijing, China), followed by a PBS rinse. Staining results were observed under a microscope, with carbohydrates and glycoproteins appearing magenta, enabling comparative analysis.

### 2.8. Flow Cytometry Analysis

RA-treated and untreated PGC were collected, with approximately 1 × 10^6^ cells per tube. The cells were centrifuged at 1200 rpm for 5 min and washed twice with PBS. Subsequently, 4 mL of ice-cold (−20 °C) ethanol was gradually added to the 1 mL cell suspension during high-speed vortexing to ensure thorough mixing. The cells were then fixed at −20 °C for 30 min, centrifuged, and washed three times with PBS. The PGC suspension was treated with RNase A at a concentration of 100 µg/mL and incubated at 37 °C for 20 min. Propidium iodide staining solution was added, and after 15 min in the dark, the samples were analyzed using flow cytometry.

### 2.9. RNA-Seq

RNA library for RNA-seq was prepared as follows: mRNA was purified from total RNA using polyT and then fragmented into 300~350 bp fragments. The first strand of cDNA was reverse-transcribed using fragmented RNA and dNTPs and second-strand cDNA synthesis was subsequently performed. Remaining overhangs of double-strand cDNA were converted into blunt ends via exonuclease/polymerase activities. After adenylation of 3′ ends of DNA fragments, sequencing adaptors were ligated to the cDNA and the library fragments were purified. The template was enriched by PCR, and the PCR product was purified to obtain the final library.

### 2.10. ATAC-Seq

Cells were collected at room temperature and counted using the Countstar Rigel S2 (Shanghai Ruiyu, FL20447, Shanghai, China). The required number of cells were lyzed for 5 min, and the nuclei were collected. The nuclei were then resuspended in a transposition reaction system containing Tn5 transposase for incubation and DNA purification. The products were subsequently amplified with the incorporation of specific indices. The amplification cycle consisted of: 3 min at 72 °C, 1 min of pre-denaturation at 98 °C, 10 s of denaturation at 98 °C, 25 s of annealing at 60 °C, and 25 s of extension at 72 °C, for a total of 13–15 cycles, followed by a final extension at 72 °C for 5 min. Library fragments of approximately 200–700 bp were selected using magnetic beads. The concentration of the library was measured using a Qubit (Thermo, Qubit 3.0, Waltham, MA, USA), and the fragment quality was assessed using the Bioanalyzer 2100 (Agilent, Santa Clara, CA, USA). Paired-end 150 bp (PE150) sequencing was performed on the Illumina NovaSeq XP platform according to standard operating procedures.

### 2.11. Statistical Methods

In this study, each experiment was conducted at least three times. Quantitative RT-PCR data were analyzed using the ΔΔCT method to calculate the relative expression levels of target genes. The expression levels of genes of interest were normalized to the reference gene β-action. Statistical differences between the experimental and control groups were determined using Student‘s *t*-test. A *p*-value of <0.05 was considered statistically significant. All data are presented as mean ± standard deviation (SD). The statistical analysis was performed using GraphPad Prism (version 9.5.0.).

## 3. Results

### 3.1. Identification of In Vitro Cultured PGC

In the process of isolating PGC from chicken embryos, a portion of blood was also collected from the embryos for DNA extraction, followed by sex determination using sex-specific primers. Subsequently, PGC derived from male and female embryos were cultured separately. Identification of PGC relies on specific molecular markers that are stably expressed across different developmental stages and culture conditions. Common PGC markers include *CVH*, *CDH*, *DAZL*, *NANOG*, and *POUV* (Figure 1B). The full electrophoresis gel image is provided in the Appendix A. The primers used are shown in Table 1. These molecular markers not only facilitate precise identification of PGC but also play critical roles in maintaining their undifferentiated state and functionality [16,17,18].

In addition, alkaline phosphatase activity serves as an essential indicator for assessing the differentiation status of stem cells in vitro. As stem cells, PGC possess abundant alkaline phosphatase [19], with AP staining displaying a strong positive signal in PGC (Figure 1C). DF-1 cells, which served as a negative control and showed no AP activity, further confirming the stem cell characteristics of PGCs. Furthermore, PGC are rich in glycogen granules, allowing for the detection of intracellular glycogen via PAS staining [20]. Observations revealed PAS-positive reactions in the cytoplasm, appearing as a characteristic purplish-red color (Figure 1D), indicating high metabolic activity and further validating their stem cell properties. In contrast, DF-1 cells, used as a negative control, showed no PAS staining.

Moreover, PGC express embryo-stage-specific antigens, such as SSEA1 and epithelial membrane antigen EMA1, which are critical markers in confirming their stem cell identity [21,22]. Staining results for these antigens are shown in Figure 1E,F. In addition, secondary antibodies were used in control experiments to demonstrate the specificity of the antibodies. Together, these characterizations underscore the essential role of PGC in reproductive development and highlight their potential as stem cells. Additionally, under in vitro culture conditions, PGC demonstrated robust proliferation and the ability to maintain their undifferentiated state, indicating that the in vitro culture system successfully mimics the in vivo environment. This system also offers a viable platform for studying interactions between PGC and exogenous RA, thereby enabling investigation into the mechanistic role of RA in PGC regulation.

### 3.2. In Vitro Induction of Meiosis in PGC by RA Treatment

To evaluate the effect of exogenously added RA on the cell cycle progression of PGC, we performed flow cytometry to analyze cell cycle distribution across experimental groups. Comparison of the cell cycle phases between female control and RA-treated groups revealed a significant increase in the proportion of cells in the G1 phase from 13.1% to 47.2% following RA treatment, and the proportion of cells in the G2 phase decreased from 54.9% to 24.1% (Figure 2A,B). Similarly, in the male control and RA-treated groups, the cell cycle showed a consistent shift, with an increased G1 phase proportion and a reduction in the G2 phase population (Figure 2C,D). These results suggest that RA treatment may induce G1 phase arrest in PGC, thereby reducing the percentage of cells progressing to the G2 phase.

In addition, Giemsa staining was performed on PGCs from both male and female control and RA-treated groups. Comparative analysis of the staining results revealed morphological changes and chromosomal alterations in a subset of PGC following RA treatment, indicating that these PGCs may have initiated meiosis (Figure 3).

### 3.3. RNA-Seq Data

Transcriptome sequencing was conducted on 12 PGC samples from both male and female groups, divided into RA-treated and control groups (*n* = 6 per condition, with three male and three female samples each). After RNA extraction, library preparation, and sequencing, the data underwent quality assessment, confirming high-quality sequencing metrics. Following filtration, a total of 6681.8 million clean reads were obtained, with GC content ranging from 48% to 48.60%. Sequencing quality was assessed using Q20 and Q30 thresholds (which represent the percentages of bases with sequencing error rates below 1% and 0.1%, respectively), with Q20 values exceeding 98% and Q30 values around 95%, indicating data of high quality suitable for downstream analyses.

Alignment of clean reads to the chicken reference genome GRCg7b showed that over 93% of reads could be mapped to the genome. Principal Component Analysis (PCA) indicated that PCA1 accounted for 97.2% of variance across samples (Figure 4A), underscoring its role as the primary distinguishing factor between sample groups. M1 and F1 groups (RA-treated) were distinctly separated from M0 and F0 groups (Control), suggesting significant differences between RA-treated and control PGC. Additionally, a Pearson correlation analysis of gene expression levels across samples revealed high correlations, with all correlation coefficients above 0.96 (Figure 4B).

Analysis of differentially expressed genes (DEG) between the RA-treated male (M1) and control male (M0) groups revealed distinct transcriptional profiles. Using stringent filtering criteria of |log2(fold change)| > 1 and a q value < 0.05, a total of 1359 DEG were identified. Among these, 545 genes were upregulated, and 814 genes showed downregulation. Similarly, a comparison between the RA-treated female (F1) and control female (F0) groups identified 1561 DEG, with 699 genes upregulated and 862 downregulated. The differential expression distribution across groups is visually summarized in a volcano plot (Figure 5A), which highlights the significant expression changes between these conditions. The Venn diagram (Figure 5B) was used to illustrate the overlap in DEG between the male and female comparisons, showing that 728 DEG were shared across both RA-treated groups. This overlap suggests that RA exposure induces some common regulatory effects on gene expression in PGC across sexes.

The GO enrichment analysis of DEG revealed significant enrichment across multiple biological processes, cellular components, and molecular function categories (Figure 5C,E). Notably, DEG in both male and female comparison groups were highly enriched in biological processes such as multicellular organismal process (GO:0007275), multicellular organism development (GO:0032501), and system development (GO:0048731). These results imply that the associated genes may contribute to the development of reproductive organs and cells, including gametogenesis and cell-type specialization [23,24]. Interestingly, DEG in the M1 and M0 comparison group were particularly enriched in cell differentiation (GO:0030154) and cell developmental processes (GO:0048869), suggesting that exogenous addition of RA affected the differentiation process of male PGC [25,26]. Meanwhile, DEG were more enriched in cell motility (GO:0048870) and cell migration (GO:0016477) in the F1 and F0 comparison group, suggesting that exogenously added RA may have affected the migration and mobility of female PGC. Additionally, DEG in both comparison groups were enriched in cellular components such as the cell periphery (GO:0071944) and plasma membrane (GO:0005886), indicating significant expression changes in genes associated with cellular membranes. KEGG pathway enrichment analysis further revealed potential involvement in intracellular signaling pathways, particularly the MAPK signaling pathway and cytokine–cytokine receptor interaction [27,28], which are known to play roles in cell proliferation and differentiation (Figure 5D,F).

### 3.4. ATAC-Seq Data

To investigate the effects of exogenous RA on chromatin accessibility and transcription factor binding sites in PGC, we conducted ATAC-seq on male and female PGC, dividing the samples into RA-treated and control groups. This setup included a total of 12 samples (*n* = 6 per condition, with three male and three female samples in each group). Each ATAC-seq library showed significant read enrichment around transcription start sites, with nucleosome-free and mononucleosome fragments clearly present in the data (Figure 6A), indicating that we successfully captured regulatory regions associated with active genes. To further assess the data quality, we performed PCA across all samples. PCA results demonstrated consistent clustering among biological replicates and clear separation between treated and control groups, supporting the high quality and biological relevance of the ATAC-seq data (Figure 6B).

Differential peaks in the RA-treated and control groups were subsequently analyzed and filtered by |log2(fold change)| > 1 and q value < 0.05. In male PGC, 1903 differential peaks were observed, with 250 upregulated and 1653 downregulated peaks. Similarly, in female PGC, comparison of the RA-treated and control groups identified 2624 differential peaks, with 1033 upregulated and 1591 downregulated peaks. The differences between these groups were further visualized in volcano plots (Figure 6C,D), clearly illustrating the peak distribution across the two conditions.

For functional insights, we conducted GO enrichment analysis on differential peaks, observing similar enrichments across biological process, the cellular component, and molecular function categories for both female and male comparison groups (Figure 6E). In the biological process category, significant enrichment was observed for the cellular process (GO:0009987) and metabolic process (GO:0008152), suggesting that genes associated with these accessible chromatin regions are likely involved in active cell activities and fundamental metabolic processes. This enrichment implies that the cells may have been undergoing rapid differentiation [29,30]. KEGG pathway enrichment analysis further highlighted significant enrichment in metabolic pathways in both comparison groups (Figure 6F). This result suggests that RA treatment may have induced metabolic reprogramming within PGC, a phenomenon often recognized as a hallmark of cellular transitions, particularly in stem cells and differentiating cells [31,32,33]. These findings indicate that RA-induced chromatin changes in PGC are linked to metabolic adjustments that potentially underlie differentiation and developmental processes.

### 3.5. Integration of ATAC-Seq with RNA-Seq

To examine whether changes in chromatin accessibility from the ATAC-seq analysis correspond with gene expression shifts, we integrated the ATAC-seq and RNA-seq datasets. By intersecting the differential genes from both sequencing analyses, we identified a total of 62 DEG (Figure 7A), comprising 20 upregulated and 42 downregulated genes. Based on statistical significance, we focused on eight DEG, which are reported to potentially function in cellular differentiation processes [34,35,36]. To further elucidate the relationship between chromatin accessibility and gene expression and investigate transcription factor-mediated regulation of downstream genes, we analyzed the correlation between chromatin openness and gene expression levels. As shown in Figure 7B, chromatin accessibility and transcriptional levels of *CYP26A1* and *HNF1B* were significantly higher in the RA-treated group compared to the control group. Notably, *CYP26A1* expression was almost undetectable in the control group, but in the RA-treated group, its expression was both frequent and markedly elevated. This suggests that RA treatment enhances chromatin accessibility and transcriptional activation of these genes.

To verify the accuracy of our RNA-seq data, we performed qRT-PCR validation on the expression of eight selected genes (*CYP26A1*, *MYO1C*, *HNF1B*, *GDPD5*, *MB21D2*, *MEIS1*, *SOX5*, and *LMO1*) using total RNA isolated from male and female PGC in both control and RA-treated groups, where the primers used are shown in Table 2. The expression patterns obtained from qRT-PCR analysis were consistent with those observed in RNA-seq (Figure 7C), thereby confirming the reliability of our RNA-seq results. This consistency supports the validity of our transcriptomic findings and reinforces the robustness of the identified gene expression changes in response to RA treatment.

## 4. Discussion

In mammals, the timing of germ cell entry into meiosis is sexually dimorphic [37,38]. In embryonic mouse ovaries, germ cells initiate meiosis around 13.5 days post coitum [39], and in embryonic mouse testes, germ cells do not enter meiosis at this stage. Instead, meiosis in male germ cells is delayed until approximately 10 days postpartum, when *STRA8* expression reaches its peak [40]. The *STRA8* is essential for both oogenesis and spermatogenesis in mice, as germ cells lacking *STRA8* expression fail to initiate meiosis, thereby halting the meiotic process [41,42]. Studies have shown that in embryonic ovaries, RA synthesized by the kidneys is pivotal in triggering meiosis in germ cells [43]. In fetal testes, however, endogenous RA is degraded by the enzyme *CYP26B1*, which ensures that male germ cells delay meiosis until after birth [44,45]. These findings underscore the importance of a finely tuned balance between RA synthesis and degradation, alongside the regulation of RA-responsive genes, in facilitating normal development of the murine reproductive system. Together, these processes are essential to ensure proper germ cell maturation and the orderly initiation of meiosis.

Although previous studies have identified RA as a key factor for initiating meiosis in chicken germ cells [46,47], its regulatory effects in embryonic gonads may vary by sex. To investigate the mechanism of RA in avian reproductive development, we administered exogenous RA to male and female PGC and compared cell cycle distributions between the RA-treated and control groups. Results indicated an increase in the proportion of PGC in the G1 phase and a decrease in the G2 phase following RA treatment in both sexes, but obviously, the tendency to increase and decrease was more significant in female PGC. Based on established RA mechanisms and the supporting literature [48,49], RA promotes the transition of PGC from a proliferative to a differentiated state by regulating the expression of differentiation-related genes.

Cells typically pause in the G1 phase before initiating differentiation, allowing time for essential regulatory mechanisms to prepare the cell for specialized function [50]. This characteristic aligns with the hypothesis that RA treatment promotes the differentiation of PGC. Giemsa-staining results further showed that some PGC underwent morphological changes and chromosomal alterations after exposure to RA, suggesting that they were progressing towards meiosis. However, not all PGCs experienced this transition, suggesting that externally supplied RA may have been partially degraded, resulting in a concentration that was insufficient to induce universal meiosis in all cells.

Using RNA-seq, we analyzed the impact of exogenous RA on gene expression within male and female PGC. Our results identified thousands of DEG in both sexes following RA treatment, underscoring RA’s broad regulatory effects on gene expression in PGC. Among these DEG, we focused on the *CYP26A1* gene, previously reported to encode a cytochrome P450 enzyme crucial in RA metabolism by facilitating its degradation [51]. In RA-treated PGC from both males and females, *CYP26A1* expression was markedly upregulated. ATAC-seq data further revealed a significant increase in chromatin accessibility at the *CYP26A1* locus, suggesting that RA treatment elevates intracellular RA levels and that PGC respond with a feedback mechanism to modulate RA concentration. Additionally, *CYP26B1* was absent from the DEG, suggesting that in chicken PGC, *CYP26A1* may play a more central role in maintaining RA homeostasis.

Among the 62 intersecting DEG obtained by combined analysis of RNA-seq and ATAC-seq, we identified seven genes (*MYO1C*, *HNF1B*, *GDPD5*, *MB21D2*, *MEIS1*, *SOX5*, and *LMO1*) associated with cell differentiation or signal transduction, hinting that RA may influence PGC differentiation through these pathways. For instance, *SOX5*, a transcription factor known to regulate chondrogenesis and neural differentiation [34], may modulate PGC fate decisions by interacting with downstream targets involved in lineage specification. *HNF1B*, a key regulator in epithelial and endodermal differentiation [35], might contribute to RA-induced signaling in PGCs. Additionally, *MEIS1*, which is implicated in Hox gene regulation [36], could play a role in the spatial and temporal orchestration of PGC differentiation.

However, there was no significant upregulation of *STRA8*, a gene traditionally associated with meiotic entry, likely due to partial degradation of exogenous RA within the cells. This suggests that the intracellular RA concentration may not have reached the threshold required to trigger meiosis universally across all PGCs. Alternatively, PGCs may exhibit a specific temporal window of sensitivity to RA, beyond which the induction of meiotic genes becomes less effective.

## 5. Conclusions

In conclusion, this study demonstrates that exogenous RA effectively promotes the transition of chicken PGC from an undifferentiated state to a differentiated state, further progressing into meiosis. Through a combined analysis of RNA-seq and ATAC-seq, we identified several key genes likely involved in RA-mediated regulation of meiotic entry in PGC, including *CYP26A1*, *MYO1C*, *HNF1B*, *GDPD5*, *MB21D2*, *MEIS1*, *SOX5*, and *LMO1*. Notably, *CYP26A1* was significantly upregulated following RA treatment, along with a marked increase in chromatin accessibility at this gene’s locus, suggesting its pivotal role in RA degradation and homeostasis regulation. These findings indicate that chicken PGC possess an intrinsic ability to modulate RA levels, possibly through upregulation of *CYP26A1*, and to metabolize excess RA, thereby delaying meiotic initiation and maintaining a developmental state conducive to germ cell maturation.

Additionally, we identified that the functions of several other genes were associated with cell differentiation and signaling, suggesting that these genes may act as downstream effectors of the RA signaling pathway, facilitating the shift of PGC from proliferation to differentiation. These findings expand our understanding of RA’s mechanistic role in the development of chicken PGC and offer new molecular targets for in-depth investigation into the regulatory networks that govern germ cell differentiation and meiotic entry.

## Figures and Tables

**Figure 1 animals-15-00023-f001:**
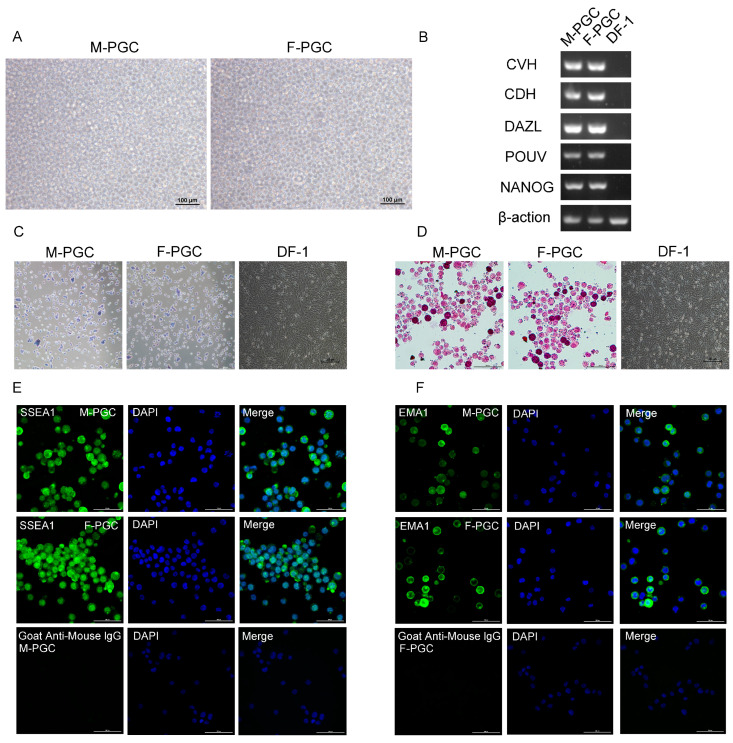
Cultivation and identification of male and female primordial germ cells (PGC): (**A**) Male and female PGC after separation of chicken embryos; (**B**) Identification of PGC marker genes, where DF-1 cells were used as a control; (**C**) Photomicrograph of male and female PGC stained for alkaline phosphatase activity; DF-1 cells as a negative control result; (**D**) Photomicrograph of male and female PGC stained with Periodic Acid-Schiff (PAS) reagent; DF-1 cells as a negative control result; (**E**) SSEA1, DAPI, and merged plots of male and female PGC; Secondary antibodies were used as control. (**F**) EMA1, DAPI, and merged plots of male and female PGC; Secondary antibodies were used as control.

**Figure 2 animals-15-00023-f002:**
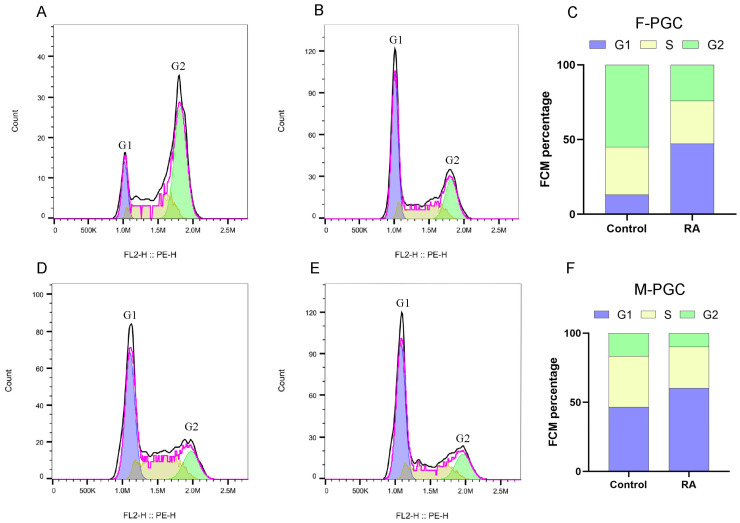
Cell cycle analysis of primordial germ cells (PGC). (**A**) The cell cycle of PGC in the female control group; (**B**) Cell cycle of PGC in the female RA-treated group; (**C**) Bar chart of the cell cycle distribution of female PGC; (**D**) The cell cycle of PGC in the male control group; (**E**) Cell cycle of PGC in the male RA-treated group; (**F**) Bar chart of the cell cycle distribution of male PGC.

**Figure 3 animals-15-00023-f003:**
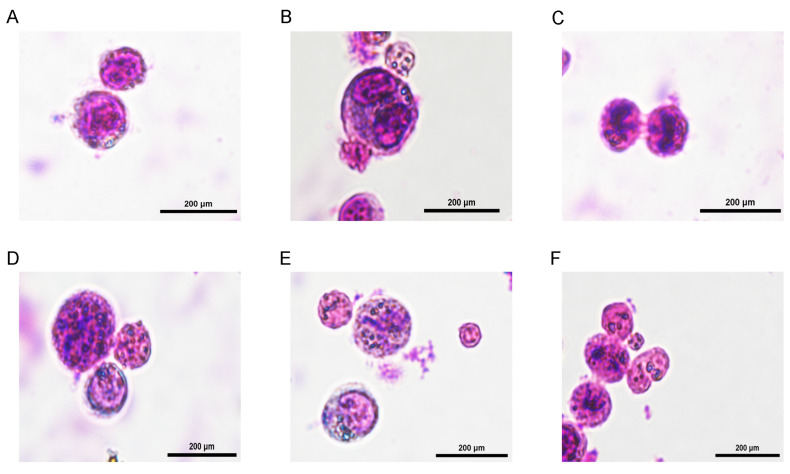
Giemsa-stained image of primordial germ cells (PGC). (**A**) Morphology of normal PGC; (**B**) PGC entering the mitotic state; (**C**) PGC following mitotic; (**D**) PGC at the leptotene stage of meiosis; (**E**) PGC at the zygotene stage of meiosis; (**F**) PGC following meiosis.

**Figure 4 animals-15-00023-f004:**
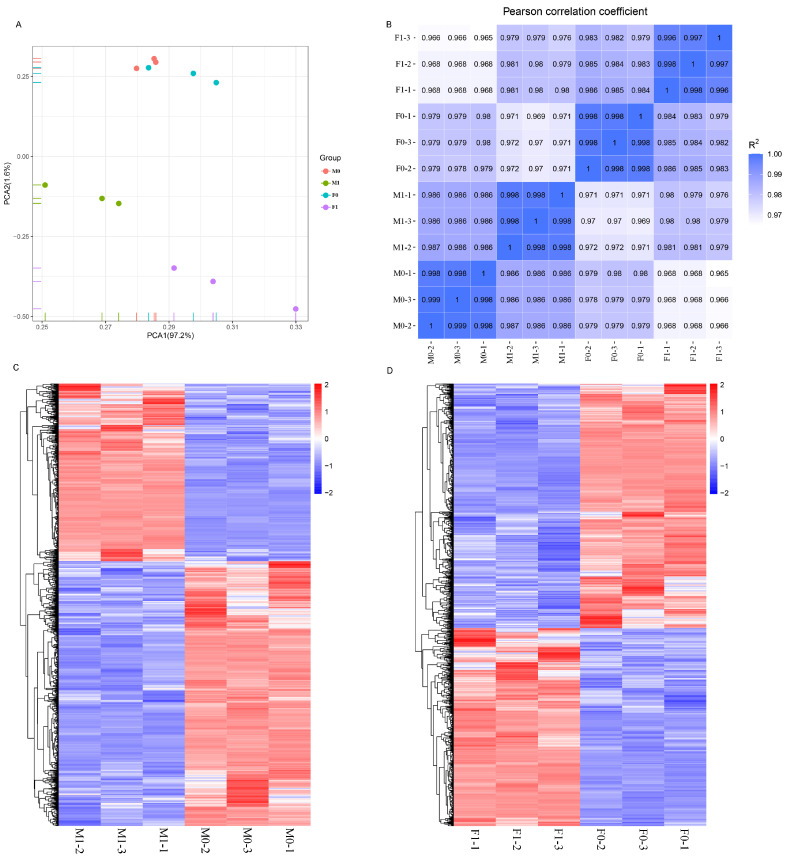
Sample relationship analysis in RNA-seq. (**A**) Principal component analysis plot between samples; (**B**) principal component analysis plot between samples; (**C**) heatmap of DEG in control and RA-treated groups in male; (**D**) heatmap of DEG in control and RA-treated groups in female.

**Figure 5 animals-15-00023-f005:**
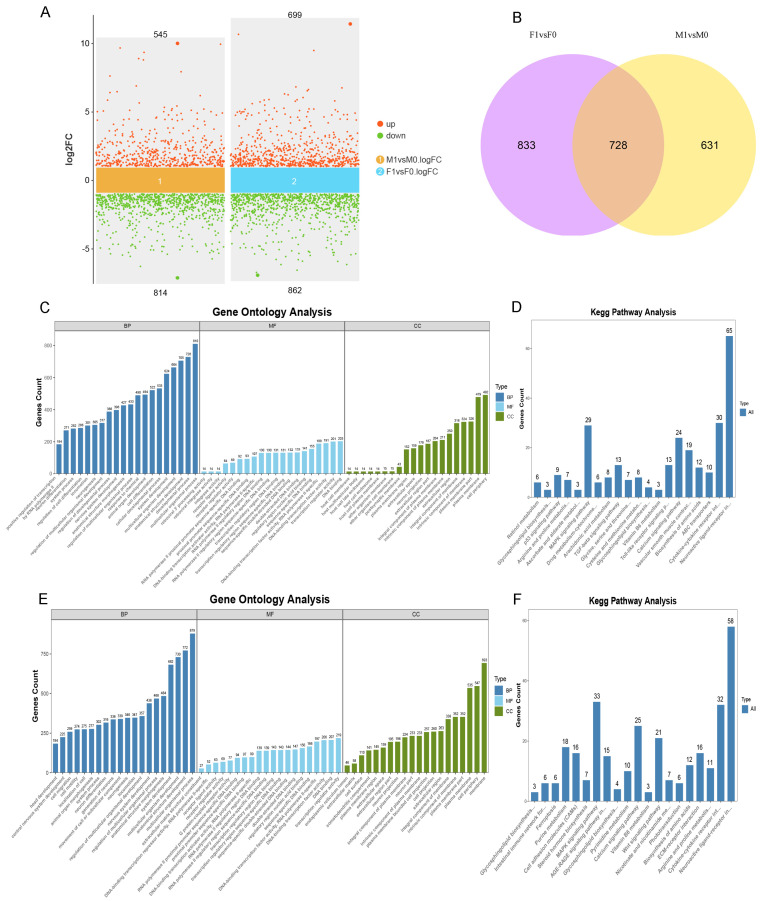
Analyses of the differentially expressed genes (DEG). (**A**) Volcano plot of DEG. M1 and F1 represent male and female RA treatment groups; M0 and F0 represent control groups of males and females. The screening threshold for DEG was |log2 FC| ≥ 1 and q value < 0.05; (**B**) Venn diagram showing intersection of DEG in male and female PGC comparison groups; (**C**) annotation of GO functions in M1vsM0 group DEG; (**D**) KEGG pathway enrichment analysis in M1vsM0 group DEG; (**E**) annotation of GO functions in F1vsF0 group DEG; (**F**) KEGG pathway enrichment analysis in F1vsF0 group DEG.

**Figure 6 animals-15-00023-f006:**
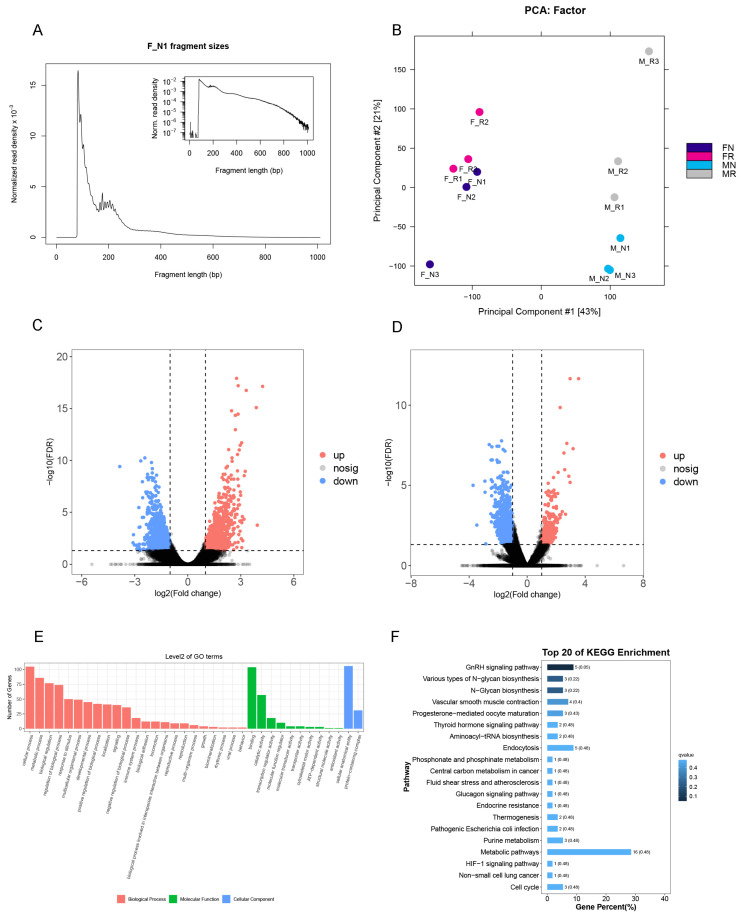
Analyses of the ATAC-seq. (**A**) Fragment length distribution map; (**B**) Principal component analysis plot. FN and MN are female and male control groups; FR and MR represent female and male RA treatment groups; (**C**) volcano map showing the different peaks between FN and FR; (**D**) volcano map showing the different peaks between MN and MR; (**E**) annotation of GO functions in different peaks between FN and FR; (**F**) KEGG pathway-enrichment analysis in different peaks between FN and FR.

**Figure 7 animals-15-00023-f007:**
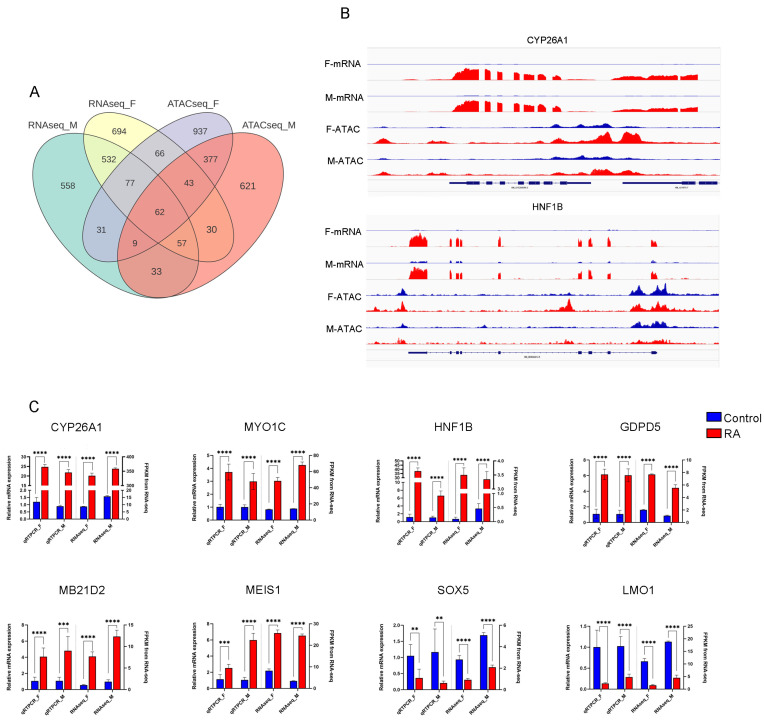
RNA-seq and ATAC-seq for combined analysis; (**A**) intersection of different mRNAs and different peaks corresponding to the target genes, F and M represent female and male comparison groups; (**B**) peak plots showing the relationship between genes in open regions of chromatin and gene expression, where the blue peak is the control group and the red peak is the RA treatment group; (**C**) validation of RNA-seq data by qRT-PCR. The left axis represents the relative expression levels determined by qRT-PCR and the right axis represents the FPKM values determined by RNA-seq. All data represent the mean of three biological replicates. The error line represents the standard error of the three replicates, and all data were normalized. ** *p* < 0.01, *** *p* < 0.001, **** *p* < 0.001.

**Table 1 animals-15-00023-t001:** Primers for identification of PGC marker genes.

Name	Primer Sequences	Product Length (bp)
CVH-F	5′-CCTTGCAGCCTTTCTTTGTC-3′	145
CVH-R	5′-GCCTCTTGATGCTACCGAAG-3′	
CDH-F	5′-CAACCGGACCAATAAGATGG-3′	182
CDH-R	5′- TGAGCGTGTCCTCATACAGG-3′	
DAZL-F	5′-TCCCAGAGCCCACACAGGATG-3′	160
DAZL-R	5′-AAGTGATGCGCCCTCCTCTC-3′	
NANOG-F	5′-CAGCAGACCTCTCCTTGACC -3′	187
NANOG-R	5′-TTCCTTGTCCCATCTCACC-3′	
POUV-F	5′-GTTGTCCGGGTCTGGTTCT-3′	189
POUV-R	5′-GTGGAAAGGTGGCATGTAGAC-3′	
β-action-F	5′-CCAGCCATCTTTCTTGGGTA-3′	141
β-action-R	5′-ATGCCAGGGTACATTGTGGT-3′	

**Table 2 animals-15-00023-t002:** Primers used to validate RNA-seq accuracy.

Name	Primer Sequences	Product Length (bp)
CYP26A1-F	5′-TACCGACAAGGACGAGTTCA-3′	199
CYP26A1-R	5′-CATTGTAGGAGGTCCATTTAGC-3′	
MYO1C-F	5′-CTCATCTGGATACAGCAGCTT-3′	114
MYO1C-R	5′- TTGGCTGCATGGCTGTGATA-3′	
HNF1B-F	5′-CGGGCAGAAGTGGACAGGA-3′	131
HNF1B-R	5′-AGGTGCGACTGGTTGAGGC-3′	
GDPD5-F	5′-GGGACGATGAATGGGAGA-3′	210
GDPD5-R	5′-GGGATGAGATGGTGAGGG-3′	
MB21D2-F	5′-TGGTCAACAAGATGTGCCCTAA-3′	205
MB21D2-R	5′-ATGCTGGTGGTGCTGCCTC-3′	
MEIS1-F	5′-ATGCCATCTACGGACACC-3′	129
MEIS1-R	5′-CATTGAAAGACTCGGAGGA-3′	
SOX5-F	5′-TGTGGGCTAAAGATGAAC-3′	240
SOX5-R	5′-ATTCGCCAATGCGTAATT-3′	
LMO1-F	5′-TGCCTTTGAGATGGTGATG-3′	170
LMO1-R	5′-ACTCGAAGCTCCCGTTCA-3′	

## Data Availability

The data presented in this study are available on request from the corresponding author. The data are not publicly available due to privacy restrictions and the long extension of datasets.

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
