# Peer review of "RNA-Seq and ATAC-Seq Reveal CYP26A1-Mediated Regulation of Retinoic Acid-Induced Meiosis in Chicken Primordial Germ Cells"

_animals, 2024, doi:10.3390/ani15010023_

Round 1

Reviewer 1 Report

Comments and Suggestions for Authors

In the article, Zhaochuan Wang and colleagues investigate the role of Retinoic Acid (RA) in meiosis of chicken primordial germ cells (PGCs). The study demonstrates that exogenous RA treatment alters the cell cycle distribution by increasing the proportion of cells in the G1 phase and reducing those in the G2 phase. Through the comparison of ATAC-seq and RNA-seq data, the authors identify the CYP26A1 gene as playing a critical role in RA-induced meiosis in chicken PGCs. This work presents a novel and compelling approach to understanding the molecular mechanisms underlying RA-mediated meiosis in chicken PGCs.

Below are my review comments:

In the Materials and Methods section, please ensure that all reagents are accompanied by their respective product numbers and manufacturers.

In lines 119-125, the authors describe the use of an in vitro feeder-free culture system with chicken PGCs to explore the mechanisms by which RA induces PGC entry into meiosis. In this context, could the authors clarify the necessity of preparing "feeder cells"?

In lines 182-206, it is recommended that the raw data from RNA-seq and ATAC-seq be deposited in a publicly accessible repository, such as the NCBI Sequence Read Archive (SRA), to ensure transparency and reproducibility.

In Table 1, which lists primers for identifying PGC marker genes, it would be helpful to include the length of the PCR products for each primer set.

In Figure 1E/F, the images are not fully in focus and lack sufficient clarity. Improving the resolution would enhance their quality.

In Figure 4B, the image is not clear and should be improved to ensure better resolution and visibility.

In Figure 5C-F, the image lacks clarity. It is recommended to provide higher quality images to improve their readability and details.

Table 2 lists the primers used to verify the accuracy of RNA-seq, and it would be helpful if the length of the PCR product for each primer was indicated.

Reviewer 2 Report

Comments and Suggestions for Authors

In the manuscript by Wang et al., Titled 'RNA-seq and ATAC-seq Reveal CYP26A1-Mediated Regulation of Retinoic Acid-Induced Meiosis in Chicken Primordial Germ Cells', the authors have validated the isolated chicken PGC by performing staining for AP activity, PAS staining and PGC antigen staining. Further they have analyzed the effect of RA treatment on cell cycle progression of PGC and conclude that RA treatment may induce G1 phase arrest. The authors perform very clean RNA seq and ATAC seq analysis and further integrate the two data sets to identify 62 intersecting genes including 7 genes involved in PGC differentiation. While the results presented here majorly support the conclusions, there are a few concerns that need to be addressed including experimental techniques, additional figures and elaboration of the discussion section. Following are the detail comments.

1.        Line213. Negative control for Figure1C and 1D is essential for concluding about the AP activity and PAS staining results. I would suggest using a non-PGC cell line as negative control here.

2.        Line213. Negative control for Antibody staining’s (Figure 1E and 1F) would help appreciate the PGC marker staining. A control cell line would be ideal and  at the least a secondary Ab control would good to have here.

3.        Line213. Figure 1F. M-EMA1 staining looks slightly brighter that F-EMA1 staining. However, it looks opposite in the supplementary. An explaination for this is required.

4.        Line213. Figure1C. Are all the cells here PGC or the one that have clustered and show a darker stain. If subset of the cells are PGCs they should be marked by arrows for ease of understating.

5.        Line240. The results from the Giemsa staining can be detailed in text and quantitative data. The figure 3 nicely describes the various stages but given the point the author is trying to make it looks incomplete and should probably include quantitative data for control and RA groups.

6.         Line265. The figure legend 4B needs to be corrected.

7.        It would be nice if the authors could elaborate on possible mechanism playing a role during RA induction based on the DEG obtained in both data sets in the discussion section. In my opinion this would be add a lot of value in terms of potential for future research.

8.   It would have been interesting if the Authors could have tested few top candidates from their data sets by performing some experiments shown in Figure1.

Reviewer 3 Report

Comments and Suggestions for Authors

Introduction

-  It should review on the methods between ATAC-seq and RNA-seq.

Line 48 : What is the X-stage embryo?

M & M

Line 98 : What is EP tube?

Line 114, 116 : change 1 umol/L to µM

Results

Line 321 : What is the meaning of q<0.05

Round 2

Reviewer 2 Report

Comments and Suggestions for Authors

While the points raised in the previous version have been addressed to some extent, I have the following suggestions on the revised version.

1. Line231:- Figure1. The negative control images can be placed in the same figure panel as the test (PGC) images. I think that having negative control and test images side by side makes a huge impact on the reader in terms of appreciating the result.

2. Line 261-264:- In my opinion, the comparative analysis of Giemsa staining, if not quantifiable, does not add any significant information to the manuscript. I would suggest trying alternate methods of quantifying the Giemsa staining. 
